# Copula-Based Bivariate Flood Risk Assessment on Tarbela Dam, Pakistan

**Saba Naz [1],\*, Muhammad Ahsanuddin [2], Syed Inayatullah [1] 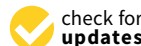, Tanveer Ahmed Siddiqi [1] and Muhammad Imtiaz [1]**

1   Department of Mathematics, University of Karachi, Karachi 75270, Pakistan
2   Department of Economics, University of Karachi, Karachi 75270, Pakistan
\*   Correspondence: sanaz@uok.edu.pk

**Abstract:** Flooding from the Indus river and its tributaries has regularly influenced the region of Pakistan. Therefore, in order to limit the misfortune brought about by these inevitable happenings, it requires taking measures to estimate the occurrence and effects of these events. The current study uses flood frequency analysis for the forecast of floods along the Indus river of Pakistan (Tarbela). The peak and volume are the characteristics of a flood that commonly depend on one another. For progressively proficient hazard investigation, a bivariate copula method is used to measure the peak and volume. A univariate analysis of flood data fails to capture the multivariate nature of these data. Copula is the most common technique used for a multivariate analysis of flood data. In this paper, four Archimedean copulas have been tried using the available information, and in light of graphical and measurable tests, the Gumbel Hougaard copula was found to be most appropriate for the data used in this paper. The primary ($T^{AND}$, $T^{OR}$), conditional and Kendall return periods have been also determined. The copula method was found to be a powerful method for the distribution of marginal variables. It also gives the Kendall return period for the multivariate analysis the consequences of flooding.

**Keywords:** flood-frequency analysis; return period; bivariate copula; tail dependence; Gumbel-Hougaard copula

---

## 1. Introduction

The environment plays an important role in our lives and has an impact on mother earth. The world is inhabited by approximately 7.7 billion people living in different regions and places. Winter, summer, spring and autumn have a bearing on people's life; the food they eat, the cloth they wear and the crops they reap all depend upon the environment. Events are planned through the information access in advance through meteorological reports according to the weather pattern. If accurate weather changes are able to be predicted, like heavy rains resulting in floods, then lives, properties, and crops can be saved.

An analysis of flood peaks just gives limited information on flood characteristics, which includes the volume, duration, and hydrograph shape of the flood, in addition to the flood peak.

A univariate display cannot catch the reliance among the factors. A portion of these investigations have thought about the reliance among flood factors, however the expectation is that all flood properties are very much described by a solitary likelihood distribution (e.g., ordinary distribution, if there should be an occurrence of bivariate typical conveyance). Previously, a flood occasion was considered as a multivariate occasion and inferred the relationship between flood factors: The peak, volume and duration utilizing an incomplete procedure technique by [1,2]. Lee used a Gumbel mixed model for the assessment of the seriousness of floods utilizing a comparison among bivariate and univariate return periods [3]. Various bivariate probability conveyances have been utilized by specialists on flood factors (peak, volume and duration). For example, Yue connected the Gumbel mixed model and Gumbel

logistic model on the sets of flood factors (peak, volume and duration) in [4,5] and [6–9] respectively, the two of which are bivariate extreme value distributions type I. These two models have been created from Gumbel marginals and the correlation of these two models has been done by [10,11]. The copula capacities have turned out to be progressively mainstream and as often as possible utilized factual apparatuses amid the previous couple of years. The expression, copula, has been first begat by Abe [12], which rose as a ground-breaking multivariate procedure for a productive examination of reliance structures among factors. In the accompanying discussion, brief subtleties of copula applications pertinent to the present investigation are discussed. Flood recurrence investigations have been studied by a few experts around the world [13–17]. De Michele and Salvadori first used copula in the field of hydrology [18] and from that point onward, various researchers have used the same approach for analysis [14,19]. Many other scientists [14,20,21] argued that the suspicion of flood factors having a similar sort of marginal probability distribution has been doubtful, and they compared copula-based bivariate distributions and the Gumbel mixed and Box-Cox transformed normal distribution. Klein connected copulas for the estimation of bivariate probabilities of produced hydrological situations for the hazard examination of a flood control framework [22]. Ben Aissia contemplated the multivariate conduct of flood arrangement with Archimedean copulas [23]. The strategy is better than multivariate distributions, since it separates the process of fitting the marginal distribution to variables from modeling the dependence among them. Ten bivariate copulas from three diverse copula families have been tried on flood information of the Sava River in Slovenia by [24]. The Gumbel-Hougaard copula has been considered as the most reasonable one for the pair's peak and volume of the flood, and the peak and duration factors, while the student's t copula was observed to be the most suitable one for the peak and duration pair of factors. [15] broke down the three flood factors, peak, volume and duration, and at the same time, utilizing a trivariate copula. For the best choice among Clayton, Gumbel– Hougaard, Frank and the student's t copula has been made based on the goodness-of-fit and tail-dependence tests and bolstered by a graphical investigation. It has been inferred that the student's t copula was the best fit for their information. As of late, another system's entropy copula implied for a multivariate examination of commonly depending factors was tried on three flood factors ($P$, $V$ and $D$) at two distinct stations of China [25]. In a study on the effect of calamitous weather on the crop insurance industry, Vergara et al. (2008) observed that 93% of crop damage is directly linked to unfavorable weather. Jones et al. (2000) are of the opinion that modeling of many variable quantities would enable farmers to make good decisions for decreasing their risk to weather exposure. Two important variables in relation to weather are rainfall and temperature that have a big effect on crop harvest (Runge 1968; Abbate et al., 2004; Calderini et al., 1999; Medori et al., 2012). In financial economics, copulas have been utilized for a long period of time (Malevergne & Sornette 2003; Patton 2009; Genest et al., 2009). In order to select the best copula, some techniques are available out of which one of the techniques to select the best copula is based on distance measures relating to the distributions of the copulas and the observed distribution of the data (Gregoire et al., 2008; Kole et al., 2007).

In this paper, a bivariate flood frequency analysis based on copula on the peak flow and volume of the flood in Tarbela dam has been performed. The Archimedean copulas, namely, Clayton copula, Frank copula, Gumbel Hougaard copula and Joe copula, have been tested. The copulas have been compared through the goodness-of-fit statistics, upper tail-dependence coefficients and graphical analysis. In order to understand the risk of occurrences of flood, the primary, conditional and Kendall return periods have been also calculated for a better understanding of the river flow in Pakistan.

## 2. Methodology

### 2.1. Copula Theory

Copula was first time used in a statistical point of view by [12] and a significant number of the essential outcomes on copula can be followed by the early work of Wassily Hoeffding [26,27].

### 2.1.1. Sklar's Theorem

The Sklar's theorem states the idea of a joint cumulative function $F_{X1X2}$ of any couple $(x, y)$ of continuous random variables at $(x, y)$ is given as:

$$F_{XY}(x, y) = C(F_X(x), F_Y(y)) \qquad x, y \in \mathbb{R} \tag{1}$$

where $F(x) = s$ and $F(y) = t$ are the marginal distribution function. If $F(x)$ and $F(y)$ are continuous, then the copula function is uniquely defined by the equation:

$$C(s, t) = F_{XY}\big(F^{-1}(s), F^{-1}(t)\big), \qquad 0 \le s, t \le 1 \tag{2}$$

where the inverse marginal distribution functions are $F^{-1}(s)$ and $F^{-1}(t)$. Conversely, it can be established that if $C$ is a copula and $F_X(x)$ and $F_Y(y)$ are the marginal distribution functions, then the function $F_{XY}(x, y)$ (as defined by Equation (1)) is a bivariate distribution function with marginal distribution functions $F_X(x)$ and $F_Y(y)$. Furthermore, if $F_X(x)$ and $F_Y(y)$ are continuous, then $C$ is unique [27].

### 2.1.2. Copula Function

Mathematically, a bivariate copula $C$ can be defined as a function that takes random variables as input marginal distribution functions and gives a joint distribution of these variables.

$C : [0, 1] \times [0, 1] \to [0, 1]$, subject to the conditions that

- ➢ $C(1, s) = C(s, 1) = s$,
- ➢ $C(0, s) = C(s, 0) = 0$ and
- ➢ $C(s_1, s_2) + C(t_1, t_2) - C(s_1, s_2) - C(t_1, t_2) \ge 0$,

whenever $s_1 \ge t_1$, $s_2 \ge t_2$, where $s$, $t$, $s_1$, $s_2$, $t_1$, $t_2 \in [0, 1]$.

A bivariate distribution of $x_1$ and $x_2$, i.e., $F(x_1, x_2)$, can be expressed in terms of the copula function $C(u_1, u_2; \theta)$ and the marginal distributions $s_1 = F_1(x_1)$ and $s_2 = F_2(x_2)$ :

$$F(x_1, x_2) = C(F_1(x_1), F_2(x_2); \theta) = C(s_1, s_2; \theta) \tag{3}$$

where $\theta$ is the parameter of the chosen copula function, describing the dependence between $x_1$ and $x_2$.

### 2.2. Dependence and Copula Fitting

Copula represents the joint distributions of correlated variables. Therefore, it is important to analyze the dependence between the variables. A graphical analysis of dependence can be done using a scatterplot of the data, a scatterplot of standardized ranks of the variables, Chi-plot and K-plot. Chi-plot is the plot of a rank-based measure of the location of each of the observations versus a measure of the Chi-squared test of independence. The dependence is positive (negative), if the transformed data are scattered above (below) the region defined by the confidence interval of the Chi-plot [28]. The K-plot proposed by [29] is the plot between the order statistics of the data and the values of these statistics expected in case of independence. If the plotted data lies near the 45-degree line, the variables are independent. While the data has strong positive dependence if the scatter of data shows curvature above the 45-degree line. Dependence measures, namely, the Pearson's correlation coefficient, $r$, Kendall's rank correlation coefficient (or Kendall's tau), $\tau$ and Spearman's rho, $\rho$ can be computed in order to quantify the dependence between variables. The $p$-values associated with these measures can be calculated under the null-hypothesis that the value of the association measures is 0. The statistical independence is rejected if $p$-value is less than the significance level. Out of these three measures of association, Kendall's tau is considered as the most important one in the theory of copulas, especially Archimedean copulas, where its relation with the generating function is utilized for the parameter estimation of the copula.

The inversion of Kendall's tau is one of the most common methods for the estimation of the parameter $\theta$ of the Archimedean copula, as mentioned earlier in this section. In this method, the relation shown in Table 1, between Kendall's tau and the parameter $\theta$ is employed for the estimation of $\theta$. This relation depends upon the generating function $\phi$ of the Copula function, thus different values of $\theta$ are obtained for different copula models.

$$\tau = 1 + 4 \int_0^1 \frac{\phi(t)}{\phi'(t)} dt \tag{4}$$

where $\phi$ and $\phi'$ are the generating function and derivative of the generating function of the Archimedean copula, respectively. This relation depends upon the generating function $\phi$ of the copula function, thus different values of $\theta$ are obtained for different copula models given in Table 1. The important characteristics of Archimedean families of copula, used in this paper, are described in Table 1.

**Table 1.** Copula function, generating function $\phi(t)$ and the functional relationship of Kendal ($\tau_\theta$) with the Copula parameter for selected Archimedean Copulas.

| Copula | $C_\theta(u,v)$ | $\phi(t)$ | $\theta$ Space | Kendall's Tau ($\tau_\theta$) |
|---|---|---|---|---|
| Clayton | $\left\{u^{-\theta} + v^{-\theta} - 1\right\}^{\frac{-1}{\theta}}$ | $\frac{1}{\theta}\left(t^{-\theta} - 1\right)$ | $[-1, \infty)$ | $\frac{\theta}{\theta+2}$ |
| Frank | $-\frac{1}{\theta} \ln\left\{ + \frac{(e^{-\theta u}-1)(e^{-\theta v}-1)}{e^{-\theta}-1} \right\}$ | $-ln\frac{e^{-\theta t}-1}{e^{-\theta}-1}$ | $(-\infty, \infty)$ | $1 + \frac{4}{\theta}(*D_1(\theta)-1)$ |
| Gumbel-Hougaard | $\exp\left\{ -((-\ln u)^\theta + (-\ln v)^\theta)^{\frac{1}{\theta}} \right\}$ | $(-lnt)^\theta$ | $[1, \infty)$ | $\frac{\theta-1}{\theta}$ |
| Joe | $\left\{(1-u)^\theta + (1-v)^\theta - (1-u)^\theta(1-v)^\theta\right\}^{\frac{1}{\theta}}$ | $-ln\left(1 - (1-t)^\theta\right)$ | $[1, \infty)$ | $1 + \frac{4}{D_k}(\theta)$ |

$^*$ D$_{K(x)}$ is the Debye function for any positive integer k, $D_k(x) = \frac{k}{x^k}\int_0^x \frac{t^k}{e^t-1}$ dt.

## 2.3. Goodness-of-Fit

After fitting the copula models, the goodness-of-fit of the models under consideration needs to be assessed, so as to find the most appropriate model. For graphical analysis of the fit of the models to the data, a scatter plot is used to compare random pairs generated from the model and the rank-based pseudo-observations generated from the data. In order to analyze complete picture of the fit of the model including the marginal distributions, the marginal distributions can be fitted to the random pairs obtained from the model and compared with the data through a scatterplot. The K-plot [30] is also a useful graphical tool for checking the fit of Archimedean copula functions. The Kendall function is the cumulative distribution of the copula. The K-plot compares the non-parametric estimate of the Kendall function with the parametric estimate of the Kendall function of the copula. If the plot is scattered near the 45-degree line that passes through the origin, the model is considered to be fitted well. The K-plot for the Archimedean copula is analogous to the Q-Q plot for normal distribution. Besides, graphical analyses of some more statistics are also needed to quantify the fit of the models. The two measures of goodness-of-fit—*RMSE* (root-mean square error) and *AIC* (Akaike Information Criterion)—were used in this paper. These statistics [31] compare the joint cumulative density function (CDF) calculated from the copulas with the empirical non-exceedance probabilities from the Gringorten plotting position formula [32] as used by [33] for the bivariate case, given as:

$$F_{XY}(x_i, y_i) = \frac{\sum_{m=1}^i \sum_{l=1}^i n_{ml} - 0.44}{N + 0.12}, \tag{5}$$

where $n_{ml}$ is the number of pairs $\left(x_j, y_j\right)$ for which $x_j \le x_i$ and $y_j \le y_i$ and $N$ are the total number of observations. *AIC* and *RMSE* were calculated as follows:

$$AIC = Nlog(MSE) + 2k, \quad MSE = \frac{1}{N-k} \sum_{i=1}^{N} (x_c(i) - x_o(i))^2,$$
$$RMSE = \sqrt{MSE,} \tag{6}$$

where $k$ is the number of parameters used in the model, while $x_c(i)$ and $x_o(i)$ represent the $i$th values computed from the model and calculated from the empirical formula. The model with the smallest value of *AIC* and/or *RMSE* is considered as the most appropriate one.

Besides *AIC* and *RMSE*, the rank-based versions of well-known Cram'er-von Mises statistic $S_n$ proposed by [33,34] were also used in this paper. This statistic is available in the copula package of R (Hofert et al., 2016) and its *p*-value can be calculated using the parametric bootstrap method.

### 2.4. Tail-Dependence Test

Tail dependence is the asymptotic dependence of the data in the upper and lower extreme sides. A copula model should not only give a good fit to the data, but also model dependence in the tails in an appropriate way. The failure to model the tail dependence might give wrong predictions of extreme events and the return periods. In this work, the authors are mainly interested in modeling the upper tail dependence of the data, appropriately. The upper-tail dependence coefficient for a copula *C* can be expressed as:

$$\lambda_U = \lim_{u \to 1^-} \frac{1 - C(u, u)}{1 - u} \tag{7}$$

Using the above formula, $\lambda_U$ can be estimated for each copula and can be compared with the estimate of the tail-dependence from the data. A non-parametric estimator of tail dependence proposed by Frahm et al., (2005) and used in this paper is given as:

$$\hat{\lambda}^{CFG} = 2 - 2exp\left[ \frac{1}{n} \sum_{i=1}^{n} log\left\{ \frac{\sqrt{log \frac{1}{U_i} log \frac{1}{V_i}}}{log \frac{1}{max(U_i, V_i)^2}} \right\} \right] \tag{8}$$

The above estimator has been derived under the assumptions that the data follows an extreme-value copula, but the estimator works well even if the assumption is not satisfied [35].

### 2.5. Return Periods

The main purpose of a flood-frequency analysis is to quantify the risk of the occurrence of future events. For this reason, several bivariate return periods based on copulas have been developed by various researchers. Salvadori and De Michele (2004) emphasized the use of "OR" and "AND" return periods as defined in the following equations.

$$T_{u,v}^{OR} = \frac{\mu}{1 - C(u, v)}, \qquad T_{u,v}^{AND} = \frac{\mu}{1 - u - v + C(u, v)} \tag{9}$$

where $\mu$ is the mean-inter-arrival time of two consecutive events. Apart from these unconditional return periods, certain conditional return periods are also considered useful in hydrological studies, which are given in following equations.

$$T_{U>u|V\leq v} = \frac{\mu}{1 - \frac{C(u,v)}{u}}, \qquad T_{U>u|V>v} = \frac{\mu}{1 - u} \frac{1}{1 - u - v + C(u, v)} \tag{10}$$

On the other hand, a secondary return period, also known as Kendall's return period, is defined by [36] as follows.

$$T_x^> = \frac{\mu}{1 - K_c(t)} \tag{11}$$

where $K_c$ is the Kendall's distribution function for theoretical copula function.

*2.6. Description of Study Area*

Tarbela dam, on the River Indus, is considered one of the world's largest earth filled and second largest dams in structural volume. The dam forms the Tarbela reservoir, which is 8.5 km long with the surface area of 250 square kilometers and holds 14.3 cubic kilometers of water. It is located in the region of Haripur, Hazara Division, region of Khyber Pakhtunkhua, approximately 50 km northwest of Islamabad, Pakistan (Figure 1). The dam is 148 m high over the riverbed. The sources of the flooding in the river Indus are heavy rainfall and snow melt to the river runoff from glaciers through the Himalayas.

Pakistan is a water lacking agrarian nation. In this way, the water control is executed on the water system prerequisite and the power generation is restricted to the water system necessity and flood control.

The high flow season is in the Kharif season. Therefore, the annual maximum upstream flow has been recorded in the Kharif season (i.e., 6 months from April to September) for 36 years (1977–2012).

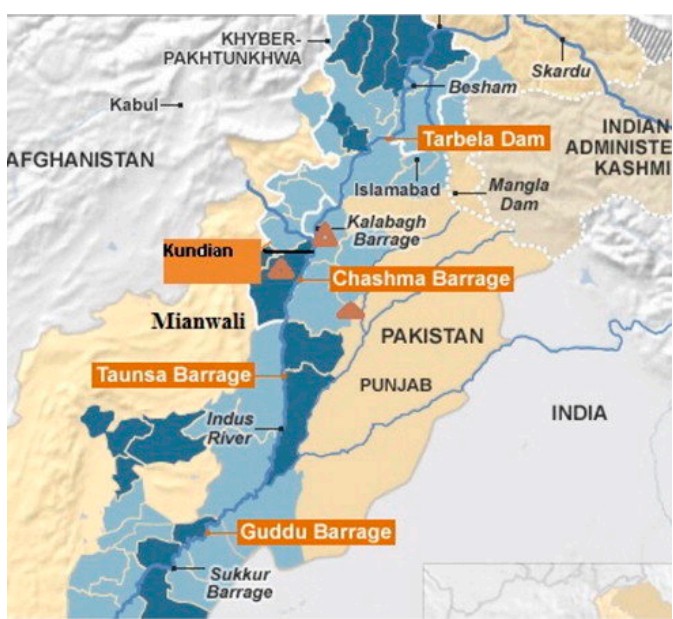

**Figure 1.** Location map of Tarbela dam.

## 3. Results and Discussion

The proposed methodology has been applied to the data obtained for Tarbela dam of Pakistan. There are two variables of a flood used, namely the peak (*P*) and volume (*V*). Table 2 depicts statistical properties of the flood variables (*P*, *V*) for the period from 1977 to 2012 (36 events). The methodology for considering the events has been published earlier [37]. The positively skewed flood parameters (*P*, *V*) of natural surroundings suggest that they can be best modelled by heavy tail distributions.

**Table 2.** Statistical characteristics of the peak flow and volume.

| Statistical Measures | Tarbela Dam | |
|---|---|---|
| | $P(\text{m}^3/\text{s})$ | $V(\text{Day-m}^3/\text{s})$ |
| Minimum | 4231 | 154,900 |
| Median | 6602 | 293,500 |
| Maximum | 15,340 | 450,000 |
| Mean | 7073 | 293,000 |
| St. Deviation | 1929.8 | 75,594.0 |
| Skewness | 2.3143 | 12,599 |
| Kurtosis | 8.4322 | 1.108 |

### 3.1. Dependence of Flood Variables (Peak and Volume)

To measure the statistical dependence between two given flood variables ($P$, $V$), the Pearson's linear correlation, and rank based correlations (Kendall and Spearmen's method) were used. The Pearson's correlation is used for examining the linear dependence between the flood variables provided that the variables are normally distributed. However, Kendall's and Spearman's method is used for ranking the variable values rather than real values and mostly, it is used in copula because they are invariant under monotonic non-linear transformations. Table 3 illustrates the dependence measures between the peak and volume of a flood with corresponding *p*-values. It can be seen that the Kendall and Spearman correlation is statistically significant (5%) for Tarbela.

**Table 3.** Dependence measures of flood peak—volume.

| Dependence Measures | Station: Tarbela Dam |
|---|---|
| Pearson's coefficient $r$ (*p*-value) | 0.358(0.031) |
| Kendall's $\tau$ (*p*-value) | 0.311(0.007) |
| Spearman's $\rho$ -(*p*-value) | 0.358(0.031) |

Marginal Distribution Analysis

Copulas can be divided into two parts for the procedure of making a joint distribution:
(1) Marginal distribution and (2) dependence structure.

For given station, marginal distributions are analyzed through the traditional single variable technique. In this study, suitable probability density functions for given data were applied namely: Generalized extreme value (GEV), Gumbel, three-parameter log-logistic, log Pearson type-III, three-parameter lognormal and Weibull distributions have been fitted to each of the two variables ($P$, $V$) (*cf.* Figure 2) using the software Easy Fit (http://www.mathwave.com/easyfit-distribution-fitting.html). The density distribution and corresponding parameters of the distribution are summarized in Table 4. It reveals that GEV is the most appropriate distribution function for modeling the peak and volume, based on the *K-S* statistic. Figure 2 also gives the same result.

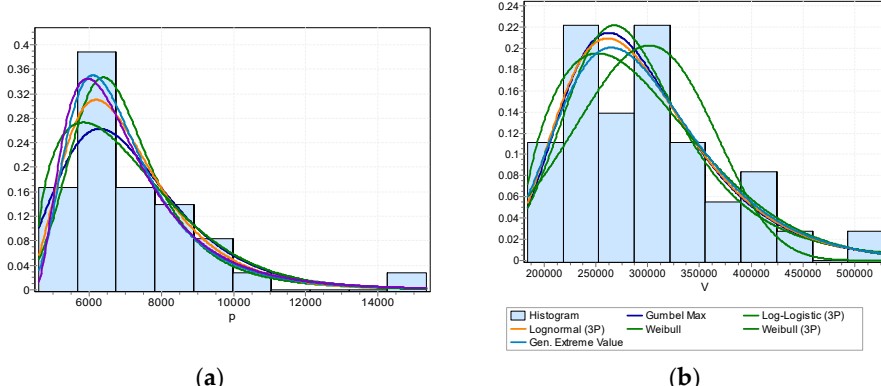

(a)　　　　　　　　　　(b)

**Figure 2.** Histogram and marginal distributions for the peak flow and volume of Tarbela dam.

**Table 4.** Marginal distributions parameters and K-S and $\chi2$ —statistics with *p*-value for the peak and volume of Tarbela dam.

| Distribution | Parameters | | K-S-Statistic (*p*-Value) | | $\chi^2$-Statistic (*p*-Value) | |
|---|---|---|---|---|---|---|
| | Peak | Volume | Peak | Volume | Peak | Volume |
| Gen. Extreme value | *Shape* = 0.1268<br>*Scale* = 1220.6<br>*Location* = 6194.8 | *Shape* = 0.1519<br>*Scale* = 68724<br>*Location* = 262770 | 0.148<br>(0.37) | 0.07906<br>(0.964) | 5.387<br>(0.249) | 0.899<br>(0.924) |
| Gumbel | *Location* = 6186.1<br>*Scale* = 1536.6 | *Location* = 56707<br>*Scale* = 260230 | 0.167<br>(0.236) | 0.1061<br>(0.773) | 5.9323<br>(0.204) | 2.3925<br>(0.663) |

**Table 4.** *Cont.*

| Distribution | Parameters | | K-S-Statistic (*p*-Value) | | $\chi^2$-Statistic (*p*-Value) | |
|---|---|---|---|---|---|---|
| | Peak | Volume | Peak | Volume | Peak | Volume |
| Log-Logistic (3P) | *Shape* = 5.306 *Scale* = 4529.9 *Location* = 2227.7 | *Shape* = 8.478 *Scale* = 3555530 *Location* = −68915 | 0.116 (0.675) | 0.0977 (0.849) | 3.582 (0.465) | 2.4461 (0.654) |
| Log-Normal (3P) | *Sigma* = 0.36208 *Mu* = 8.4098 *Gamma* = 2268.0 | *Shape* = 0.16487 *Scale* = 12.971 *Loc.* = −142830 | 0.147 (0.378) | 0.09037 (0.904) | 3.5234 (0.474) | 2.0659 (0.723) |
| Log-Pearson 3 | *Alpha* = 7.1597 *Beta* = 0.09151 *Gamma* = 8.1778 | *Alpha* = 80.337 *Beta* = −0.02833 *Gamma* = 14.83 | 0.1636 (0.260) | 0.0849 (0.937) | 7.8324 (0.097) | 2.0659 (0.723) |
| Weibull | *Shape* = 5.4169 *Scale* = 7403.1 | *Shape* = 4.6145 *Scale* = 315230 | 0.1530 (0.333) | 0.12602 (0.963) | 5.6771 (0.224) | 0.19392 (0.978) |

### 3.2. Dependence Structure between Peak and Volume Using Copula Function

The dependence between the two given flood variables (*P*, *V*) is shown through the scatter plot of standardized ranks in Figure 3a. The scatter plot shows that there is a positive relation between the peak and volume of the flood. Moreover, other rank based scatter plots, the Chi-plot and *K*-plot, are displayed in Figure 3b,c respectively. In the Chi-plot, mostly the values are located inside the region defined by the confidence intervals, which shows that positive dependency between the two variables (peak and volume) is weak. Moreover, the *K*-plot also reveals the same result. Table 4 depicts the value of dependence of the flood variables that are measured relative to *p*- values. The values of Pearson (linear correlation), Spearman's and Kendall's rank-based dependence measures verify the results obtained from the graphical methods.

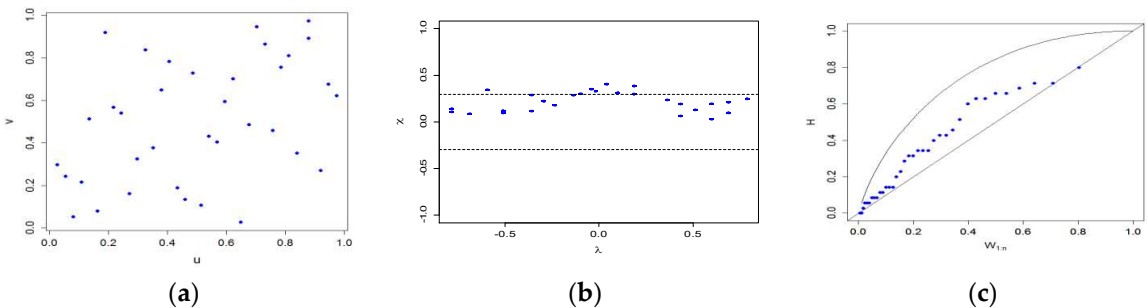

| (a) | (b) | (c) |

**Figure 3.** (**a**): Scatter plot between u and v of the peak flow and volume of Tarbela dam (**b**): Chi- plot between $\lambda$ and $\chi$ of the peak flow and volume of Tarbela dam and (**c**): K-plot between $W_{i,n}$ and H of the peak and volume of Tarbela.

### Copula Modeling

After the selection of marginal distribution of the random variables, the selection of suitable copula is performed. Therefore, four different Archimedean copulas were considered: Clayton, Frank, Gumbel-Hougaard and Joe. Copula function and parameter space, generating function and admissible function of dependence Kendall $\tau$ is shown in Table 1. The value of the Kendall's coefficient of correlation lies within the permissible ranges of these copulas [38]. The graphical comparison among the copulas under consideration is depicted in Figure 4. Ten thousand random pairs generated from the copula were compared with pairs of pseudo-observations that were weekly positively associated. Actually, the points tend to place themselves along a diagonal, while the measured values of *P* and *V* are compared with random numbers generated by the copula and marginal distributions of *P* and *V*. The Q-Q plot of the fitted copula is obtained by plotting Kendall's function of the empirical copula versus Kendall's function of the copula (under consideration). The empirical and theoretical copula

functions are compared. The solid grey line shows empirical copula and the dashed red line represents theoretical copula. The joint c.d.f function of selected copula (Gumbel- Hougaard) is given. It is difficult to decide the most appropriate copula merely using graphical analysis, since all the four copulas seem to fit well to the empirical data. The values of AIC, RMSE and $S_n$ statistic with their *p*-value, along with the parameter estimated for each of these copulas are provided. This statistic is available in the copula package of R [39] and its *p*-value can be calculated using the parametric bootstrap method. On the basis of AIC and RMSE and $S_n$ statistics, it can be concluded that Clayton, Frank and Gumbel copulas fit better to the data than Joe copula. These results are given in Table 5.

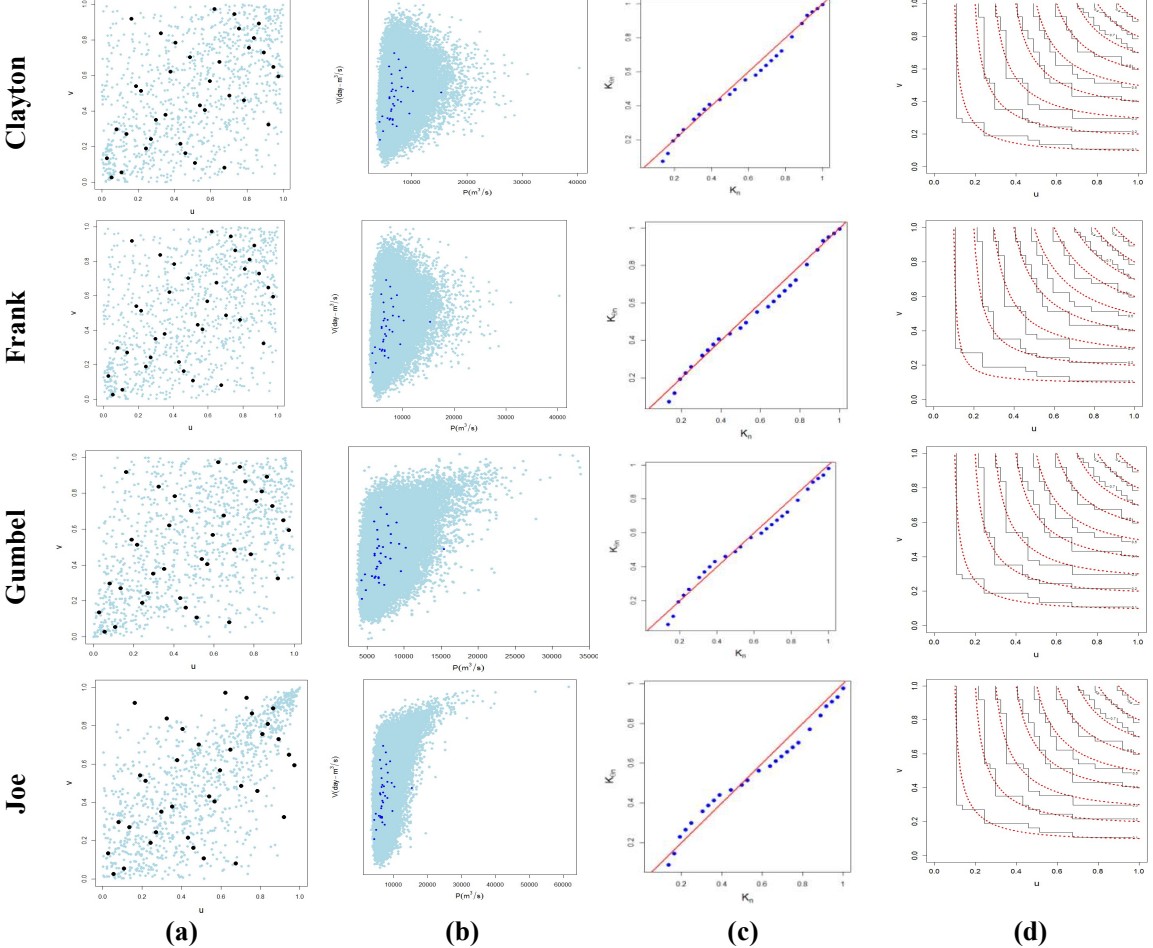

**Figure 4.** (**a**,**b**) The comparison of observed data (black dots) and 10,000 random pairs generated data (blue dots) from the copula for peak-flow and volume. (**c**) The QQ-plot of the copula (Kendall function from the data and the copula). (**d**) The comparison between empirical and theoretical copula.

**Table 5.** Estimated value of the copula parameter (θ), RMSE, AIC and Crammer-von Misses goodness-of-fit test ($S_n$) and *p*-values based on parametric bootstrap samples.

| Copula | θ | RMSE | AIC | $S_n$ | *p*-Value |
|---|---|---|---|---|---|
| Clayton | 0.9032 | 0.546 | −29.513 | 0.0264 | 0.7028 |
| Frank | 3.044 | 0.555 | −28.298 | 0.0227 | 0.898 |
| Gumbel | 1.452 | 0.562 | −27.481 | 0.029 | 0.499 |
| Joe | 1.814 | 0.5679 | −26.728 | 0.037 | 0.1803 |

### 3.3. Analysis Tail Dependence Coefficient

For the adequacy of the selected copula, the function is obtained from the tail dependence test. In order to decide the most appropriate copula between the Frank and Gumbel copulas, the tail-dependence test is performed. The current study is more interested in the upper-tail dependence of the data, since the flooding conditions occur when the values of the variables *P* and *V* are high. The empirical tail-dependence coefficient of the observed data obtained by Equation (8) $\lambda_U^{CFG}$ came out to be 0.345. The upper-tail dependence coefficient, $\lambda_U$ for the Clayton and Frank copula is 0, while $\lambda_U$ Gumbel-Hougaard copula is closer to the empirical value (cf. Table 6). This indicates that the Clayton and Frank copulas may underestimate the risk of flood, since they fail to fit to the empirical tail dependence coefficient and do not show asymptotic dependence in the upper tail of the data [40]. Gumbel-Hougaard copula is thus considered to be the most appropriate one for the given data out of the four copulas under consideration. This outcome is in accord with [16].

**Table 6.** Upper tail dependence coefficient of used the copula.

| Copulas | $\hat{\lambda}_U(\theta)$ | $\Theta$ | $\hat{\lambda}_U$ |
|---|---|---|---|
| Clayton | 0 | 0.9032 | 0 |
| Frank | 0 | 3.044 | 0 |
| Gumbel | $2 - 2^{\frac{1}{\theta}}$ | 1.452 | 0.387 |
| Joe | $2 - 2^{\frac{1}{\theta}}$ | 1.814 | 0.535 |

The c.d.f value of Gumbel Hougaard is estimated using one hundred thousand simulated values of flood variable which is plotted in Figure 5. It gives a good result because c.d.f values are tending to be 1. The c.d.f value of Gumbel-Hougaard copula is used for calculating the return periods.

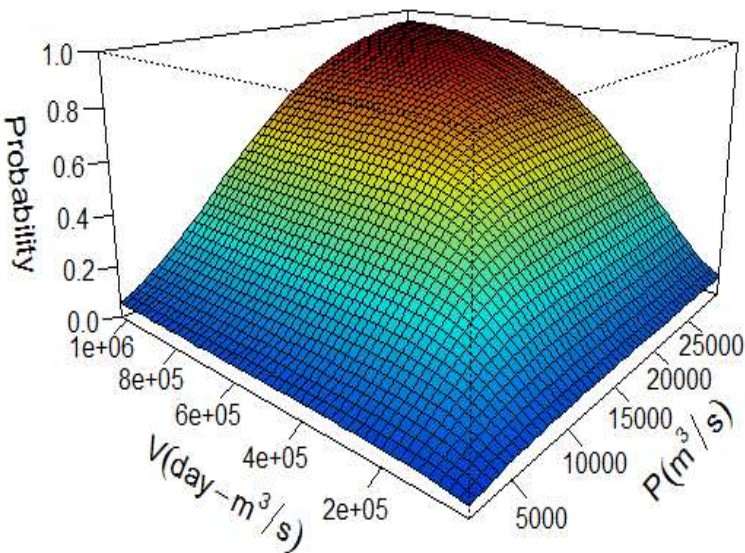

**Figure 5.** The c.d.f value of Gumbel-Hougaard Copula.

### 3.4. Primary Return Periods (T$^{AND}$, T$^{OR}$)

The c.d.f have been calculated using Equation (5) for Gumbel-Hougaard copula (cf. Figure 5). The primary return periods of flood variables (peak and volume) were calculated using Equation (9) for the Gumbel-Hougaard copula. The c.d.f is useful for the calculation of the bivariate joint return period. There may exist more than one possible combination of flood variables which is not facilitated in the univariate return period. The contour lines for particular return periods are shown in Figure 6a,b. Figure 6a shows "$T_{u,v}^{OR}$" return period, in which the flood variables (peak and volume) exceeded ($T_{u,v}^{OR}$)

outward bounds, and Figure 6b illustrates $T_{u,v}^{AND}$ return period, in which the flood variables (peak and volume) exceeded inward bounds. The $T_{u,v}^{AND}$ return period is greater than $T_{u,v}^{OR}$ return period obtained for the same peak and volumes values. For example, in the year 2010, for the annual peak flow of the corresponding volume, the primary return period for the flood event $T_{u,v}^{AND}$ is 280 years and that for $T_{u,v}^{OR}$ is 34 years which verifies Equation (8). The calculated return periods value at the median values of the peak is 6625 m³/sec flow and that of volume is 2, 93,800days-m^3/sec and the maximum value of the peak is 15,340 m^3/sec and that of volume is 449,961.63 days-m^3/sec of the hydrograph as shown in Table 2

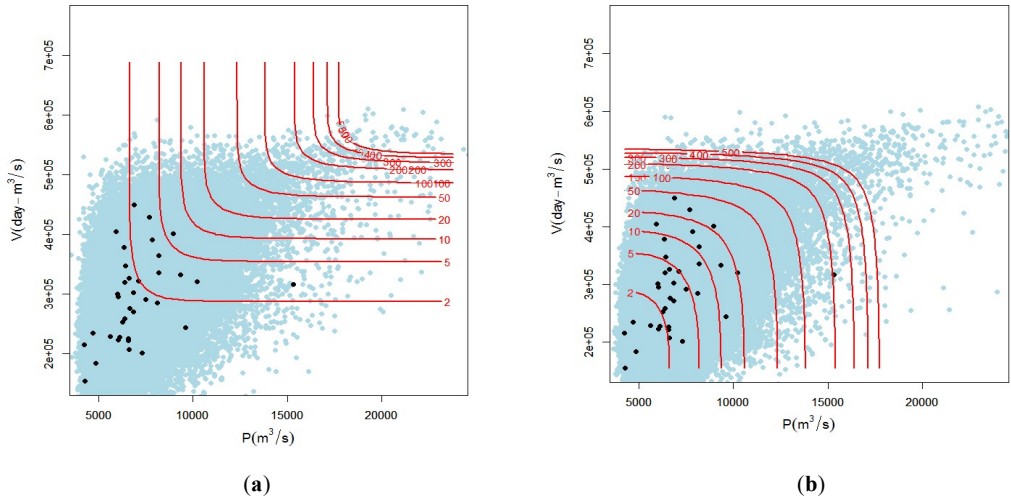

|             (a)             |             (b)             |

**Figure 6.** Primary return periods T $^{OR}$. Primary return periods T$^{AND.}$

### 3.5. Conditional Return Period ($T_{P>p|V\leq v}$ and $T_{P>p|V>v}$)

The conditional return periods of the flood peak given the volume ($T_{P>p|V>v}$) and return periods of the flood volume given peak ($T_{V>v|P>p}$) have been calculated by Equation (10) for the Gumbel-Hougaard Copula as shown in Figure 7a,b. The return periods calculated for median and maximum values of the peak flow and volume are shown in Table 7. The conditional return period $T_{V>v|P>p}$ is always bigger than $T_{U,V}^{AND}$, and the primary return period $T_{U,V}^{OR}$ is always less than $T_{V>v|P>p}$. Moreover, the differences between the maximum values of flood variables return period are considerably broader as compared to the median values.

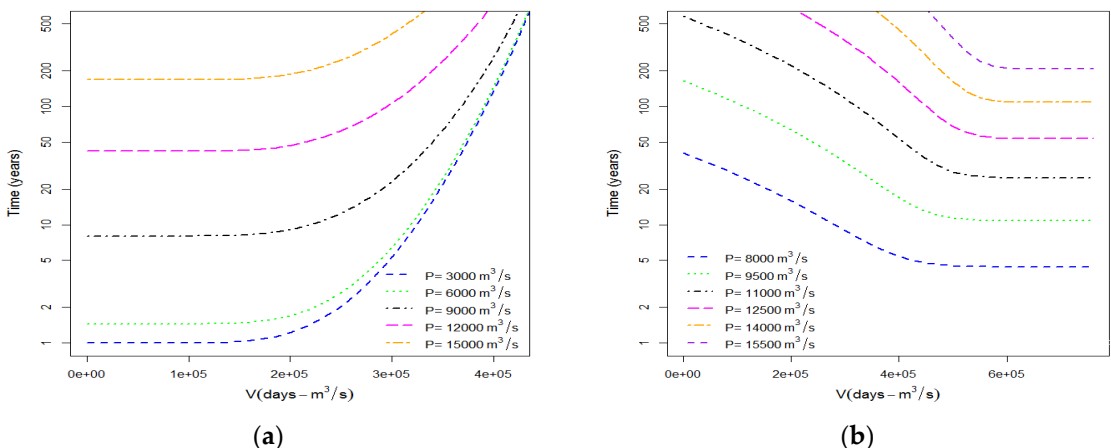

|             (a)             |             (b)             |

**Figure 7.** (**a**) Conditional return periods $T_{(V>v|P\leq p)}$, (**b**) Conditional return periods $T_{(V>v|P>p)}$.

**Table 7.** Primary and conditional return periods for copulas at the median and maximum values of *P* and *V*.

| Return Periods | Copula | Gumbel | Frank | Clayton |
|---|---|---|---|---|
| $T^{OR}$ | Median | 1.507 | 1.704 | 1.505 |
| | Max. | 34.114 | 30.936 | 30.669 |
| $T^{AND}$ | Median | 3.154 | 2.541 | 3.166 |
| | Max. | 280.48 | 1820.808 | 3737.91 |
| $T_{(U>u\|V\leq v)}$ | Median | 3.248 | 6.643 | 3.224 |
| | Max. | 41.1464 | 36.603 | 36.227 |
| $T_{(U>u\|V>v)}$ | median | 6.139 | 4.946 | 6.161 |
| | Max. | 54,585 | 353,889.807 | 726,495.046 |

*3.6. Kendall or Secondary Return Period*

The Kendall's return period was calculated by Equation (11) as described in Section 2.5 at values $t = 0.90, 0.99, 0.999, 0.9999$ (cf. Table 8). The relation between $T^{OR} \leq T^{KEN} \leq T^{AND}$ shows that the Kendall return period neither overestimates, which may increase the cost, nor underestimates that may increase the risk of failure compared to the primary return period. Figure 8a shows a relation between *t* (levels) and the Kendall return period. On the other hand, Figure 8b illustrates through a contour plot of the peak and volume, the secondary return period over 100 years. This plot shows the critical region for the value of the peak and volume.

**Table 8.** Kendall return period for different *t* values.

| *t* (Levels) | *T* (Years) | $K_{c(t)}$ | $T^{KEN}$ |
|---|---|---|---|
| 0.9 | 10 | 0.932 | 28.82 |
| 0.99 | 100 | 0.9932 | 301.84 |
| 0.999 | 1000 | 0.99932 | 3267.97 |
| 0.9999 | 10000 | 0.99932 | 37037.037 |

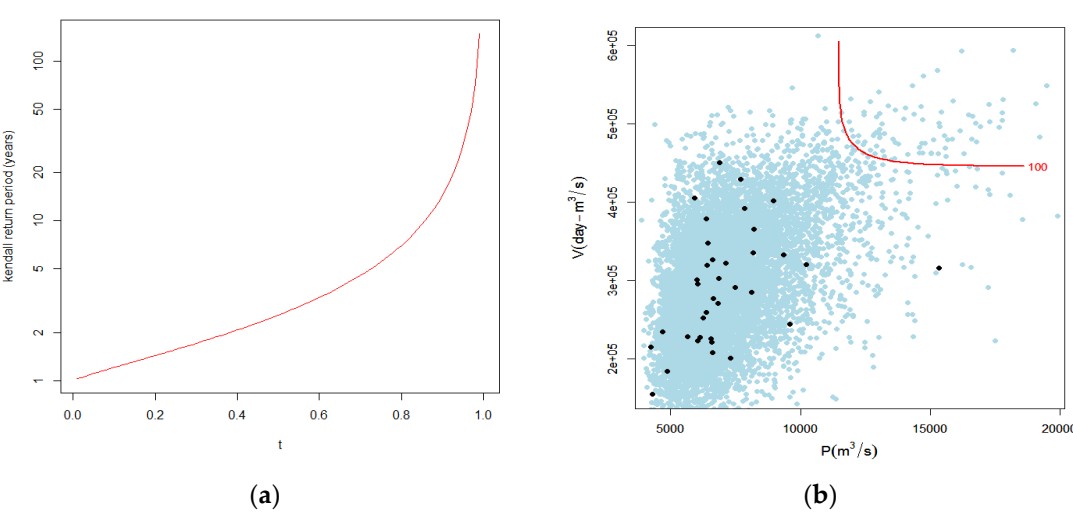

(**a**)                    (**b**)

**Figure 8.** (**a**) The plot between t (levels) and Kendall return period. (**b**) The contour plot of the peak and volume for the secondary return period over 100 years.

## 4. Conclusions

In this study, bivariate Archimedean copula-based models have been tested on hydrological data for the analysis of flood risks. It has been found that a generalized extreme value distribution

fits better as the marginal distribution to both the peak flow and flood volume of the hydrograph, whereas, Gumbel-Hougaard copula is adequate for modeling the dependency among the two variables. The return periods in a single variable analysis give overestimated results as bivariate copula modeling. For example, a return period gives many peaks for different volumes and vice versa which is useful in hydrological planning and the design of flood-protection infrastructure.

The study and research undertaken for Tarbela region during the last thirty-six years show that the risk of high flood really exists within the time span of the coming ten years. The risk of occurrence of high flood is depicted in Table 7 according to which chances lie between the ranges of 1.5 years to 34 year for $T^{or}$ i.e., the return period. In the future, the work can be replicated to the data of flood duration along with the peak flow and volume data by performing a bivariate analysis on pairs of flood variables. Furthermore, a three-way analysis of data can also be performed by using these three variables, which would give more reliable risk assessments. The extreme-value copulas and elliptical copulas were not considered in this work since this study was limited to the application of a bivariate Archimedean copula.

The risk and uncertainty analysis provides very important information to managers to make better judgments and decisions based on modeled outcomes. These results allow administrators of the dam to identify events with the potential for failure and improved understanding of the critical parameters needed for monitoring. Society has become more developed and through the cutting-edge technology available, now floods can be predicted, monitored and controlled to some extent which is in fact instrumental in saving precious lives and properties resulting in greater improvements in the world economy.

**Author Contributions:** This work was carried out in collaboration between all authors. Author S.N. designed the study and wrote the final manuscript. Author M.A. proof read and rewritten the manuscript. Authors S.I. and T.A.S. reviewed the literature. Author M.I. performed the statistical analysis, wrote the protocol, and wrote the first draft of the manuscript. All authors read and approved the final manuscript.

**Conflicts of Interest:** There is no conflict of interest between all authors.

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
