# Peer review of "Copula-Based Bivariate Flood Risk Assessment on Tarbela Dam, Pakistan"

_hydrology, doi:10.3390/hydrology6030079_

Round 1

Reviewer 1 Report

The authors used the bivariate Copula method to conduct hazard investigation of flood peak and volume in Tarbela dam of Pakistan. Overall the submission, the authors have clearly stated the research purpose and concluded a logistic outcome in your results. I think the paper is worth to be published. Thank you.

Author Response

Dear Reviewer,

Thank you for reviewing the paper and giving satisfactory report. However the other reviewer suggested some suggestions which has been duly incorporated.

With best wishes

Dr. Saba Naz

Reviewer 2 Report

This study evaluates multiple bivariate statistical procedures to quantify peak flow and flood volume recurrence intervals for flood events between 1977 and 2012 at the Tarbela Dam in Pakistan. The work focuses on the statistical methods which are described in details and with appropriate references. This study evaluates different bivariate statistical methods to quantify the recurrent interval of events based on peak discharge and volume. Multiple metrics were used to describe and compare goodness-of-fit to determine the most appropriate method. 

This is a re-submission which partially addresses previously provided suggestions and comments. I find this work important as it contributes to general knowledge by evaluating methods to quantify recurrent events and also to managers in this region, with improved information of events frequency. As of the previous submission, the discussion does not comment on the results from applying the selected model to the region. 

The use of the English language has improved but it still needs improving. 

Please find my general comments below:

Itemized comments:

L15. Should it be “Peak and volume” rather than “Peak, volume”?

L17-18. The sentence “Several attempts have been made to study flood data in different parts of the world” does not add information to the text. I suggest removing it.

L19-21. The following sentences can be combined into “Copula is the most common technique used for multivariate analysis of flood data. In this paper, four Archimedean copulas have been tried using the available information…”

L37-39. The following sentence can be re-written as “If we are somehow be able to predict accurate weather changes like heavy rains resulting in floods then we may be able to save lives, properties, and crops.”

L41-44. The following sentence can be re-written as “Analysis of flood peak just gives limited information on flood characteristics, which includes volume, duration, and hydrograph shape of flood; in addition to flood peak.”

L46. Replace “spoken” with “described”?

L56. The sentence “the two of which are bivariate sort I extraordinary esteem distributions” does not read well. Please consider rewriting it.

L64. “finished” to “studied” ??

L66. “paper” to “study”. 

L66-68. This sentence does not provide information to readers. Please consider re-writing or remove it.

L70-73. Do not start a sentence with references.

L102. The authors write “three” copulas but only list two. Please revise.

L107-116. The description of the site should be moved from the introduction section into a “study site” sub-section under the methodology section.

L118-119. These sentences can be removed.

L121-123. These sentences provide little information to the reader. Please consider re-writing.

L147. Should it be “s2” rather than “u2”?

Table 3.1 – What is the total number of events considered? How was each individual event classified into flood and non-flood? Please provide additional information.

Section 3.1 – The analysis of statistical dependence was performed by calculating indices and visual interpretation of plots. These two sections can be combined into a single section.

Figure 3.2 – please add a, b, and c notations.

L307. Replace “best” with “most appropriate”

Figure 3.3 – move the “a”, “b”, and “c” notations to the end of the sentence. Don’t start a sentence with these notations.

L329-330. The sentence “It is an important work point for bivariate joint return period.” Does not read well. Please consider re-writing it.

L372. Did the study tested “Archimedean” and “Gaussian with entropy” methods? 

L381-382. This sentence could be expanded to include information on the return period of the observed dataset.

L390-397. The sentence:

“To sum up, risk and uncertainty analysis gives very important information to decision makers to make better judgments and decisions based on the outcomes. These results allow the administrators of the dam to identify the events with a developing failure mode and understand the critical parameters that are needed to effectively monitor. Society has become more developed and through the cutting-edge technology available, now floods can be predicted, monitored and controlled to some extent which is in fact instrumental in saving precious lives and properties resulting in greater improvements in the world economy.”

Could be re-written as:

“Risk and uncertainty analysis give very important information to managers to make better judgments and decisions based on modeled outcomes. These results allow administrators of the dam to identify events with potential for failure and improved understanding of the critical parameters needed monitoring. Society has become more developed and through the cutting-edge technology available, now floods can be predicted, monitored and controlled to some extent which is in fact instrumental in saving precious lives and properties resulting in greater improvements in the world economy.”

Author Response

Dear Reviewer

I am grateful for your invaluable suggestions which are duly incorporated in the paper. The point by point details are as follows:

L15. Should it be “Peak and volume” rather than “Peak, volume”?

Ø  Suggestion incorporated

L17-18. The sentence “Several attempts have been made to study flood data in different parts of the world” does not add information to the text. I suggest removing it.

Ø  As advised, removed.

L19-21. The following sentences can be combined into “Copula is the most common technique used for multivariate analysis of flood data. In this paper, four Archimedean copulas have been tried using the available information…”

Ø  As per suggestion the sentence is now combine

L37-39. The following sentence can be re-written as “If we are somehow be able to predict accurate weather changes like heavy rains resulting in floods then we may be able to save lives, properties, and crops.”

Ø  As per suggestion the sentence is re-written.

L41-44. The following sentence can be re-written as “Analysis of flood peak just gives limited information on flood characteristics, which includes volume, duration, and hydrograph shape of flood; in addition to flood peak.”

Ø  As suggested the sentence is re-written.

L46. Replace “spoken” with “described”?

Ø  As per suggestion the word is replaced.

L56. The sentence “the two of which are bivariate sort I extraordinary esteem distributions” does not read well. Please consider rewriting it.

Ø  As identified the sentence is re-written.

L64. “finished” to “studied” ??

Ø  Done

L66. “paper” to “study”. 

Ø  Done

L66-68. This sentence does not provide information to readers. Please consider re-writing or remove it.

Ø  Sentence is re-written.

L70-73. Do not start a sentence with references.

Ø  Suggestion incorporated

L102. The authors write “three” copulas but only list two. Please revise.

Ø  Revised

L107-116. The description of the site should be moved from the introduction section into a “study site” sub-section under the methodology section.

Ø  The paragraph moved as suggested.

L118-119. These sentences can be removed.

Ø  The sentence is removed

L121-123. These sentences provide little information to the reader. Please consider re-writing.

L147. Should it be “s2” rather than “u2”?

Ø  It is replaced.

Table 3.1 – What is the total number of events considered? How was each individual event classified into flood and non-flood? Please provide additional information.

Ø  The total number of events are thirty six which is added in the paper. For further information a reference of our earlier publication is incorporated.

Section 3.1 – The analysis of statistical dependence was performed by calculating indices and visual interpretation of plots. These two sections can be combined into a single section.

Ø  For clarity I have shown this in two sections.

Figure 3.2 – please add a, b, and c notations.

Ø  Suggestion incorporated.

L307. Replace “best” with “most appropriate”

Ø  This is replaced.

Figure 3.3 – move the “a”, “b”, and “c” notations to the end of the sentence. Don’t start a sentence with these notations.

Ø  I would like to state that (a), (b), (c) and (d) are figure numbers not notations. The same pattern  is used for all the figures.  

L329-330. The sentence “It is an important work point for bivariate joint return period.” Does not read well. Please consider re-writing it.

Ø  This is re-written.

L372. Did the study tested “Archimedean” and “Gaussian with entropy” methods? 

Ø  The work tested only Archimedean method.

L381-382. This sentence could be expanded to include information on the return period of the observed dataset.

Ø  The sentence is expanded and information is included.

L390-397. The sentence:

 “To sum up, risk and uncertainty analysis gives very important information to decision makers to make better judgments and decisions based on the outcomes. These results allow the administrators of the dam to identify the events with a developing failure mode and understand the critical parameters that are needed to effectively monitor. Society has become more developed and through the cutting-edge technology available, now floods can be predicted, monitored and controlled to some extent which is in fact instrumental in saving precious lives and properties resulting in greater improvements in the world economy.”

Could be re-written as:

“Risk and uncertainty analysis give very important information to managers to make better judgments and decisions based on modeled outcomes. These results allow administrators of the dam to identify events with potential for failure and improved understanding of the critical parameters needed monitoring. Society has become more developed and through the cutting-edge technology available, now floods can be predicted, monitored and controlled to some extent which is in fact instrumental in saving precious lives and properties resulting in greater improvements in the world economy.”

Ø  This is re-written.

This manuscript is a resubmission of an earlier submission. The following is a list of the peer review reports and author responses from that submission.

Round 1

Reviewer 1 Report

This study evaluates multiple bivariate statistical procedures to quantify peak flow and flood volume recurrence intervals for flood events between 1977 and 2012 at the Tarbela Dam in Pakistan. The work focuses on the statistical methods rather than on the hydrology of the site. Individual statistical methods are described in details and with appropriate references. Multiple metrics were used to describe and compare goodness-of-fit to determine the most appropriate method. 

It is difficult to determine the main contribution of this study. It seems the main contribution is the determination of the most appropriate bivariate statistical method to quantify recurrence events in this site. However, the conclusion section fails to inform the reader what are the implications of this method selection to the region. I sincerely think there is potential in this work. For instance, quantitative and statistical modeling of past flood events can be useful to predict/prepare for future flood events. This is of significant importance in the region. However, the manuscript, in this present form, does not convey nor contains discussions of this issue.

General comments:

Abstract: 

The abstract does not read well. A complete re-write is advised.

Text: 

The entire text is difficult to read. The text needs to be revised for readability and English grammar. Additionally, the authors need to be consistent on the terminology used. The authors refer to the following terms in multiple locations in the text: “peak” and “volume of flood”, “peak flow” and “volume of the hydrograph”, “peak” and “volume”, “peak” and “volume of flood”, and “peak flow” and “flood volume”. I think using “peak flow and daily flow volume” could be an option.

Objectives:

It seems that the main objective of this study is to evaluate existing statistical methods through the use of different metrics. Is a new method being proposed? If no new method is being proposed, why this work is important? Further clarification is needed.

Figures:

It would be beneficial to the reader to see a Figure containing the location of the study site (in the world and in Pakistan) with watershed boundaries and topographic information. Does the land use/land cover change between 1977 and 2012? What are the topographic characteristics of this catchment?

It would be beneficial to the reader to see a Figure containing a hydrograph (either annually or monthly timescales) with accompanying rainfall. The authors could include a plot covering the entire study period and another “zoomed in” to one or two selected events. This would provide the reader with a sense of the hydrology in the region. Also, it would be beneficial to the reader a brief description of the weather/climate (precipitation and temperatures).

Please check figures numbering.

Figure 3.1 – plots are not of the same size.

Figure 3.3 – I am assuming the blue dots are randomly generated values (theoretical) and the black dots are observed values (empirical). This is not described in the text nor in the Figure caption. The same comment for 3.3 b.

Figure 3.6 – The text suggests one plot is conditional return period given peak flow and the other given flood volume. However, the axis description and legend in both figures are the same, Volume. Please check this.

Methodology:

Table 3.1 contains descriptive statistics for the flood events considered. How many flood events were considered? How was an event defined as flood event?

Results and discussion

Could the authors mention something about how this model can be used to describe recurrent intervals of past flood events? Is there any noticeable trend? Is there the effect of season and/or climatic factors? How about future events? Is there higher/lower probability for them to happen? It seems the study could be strengthen by including discussions relevant to action agencies.

Conclusions:

The conclusions could be stronger. Given the selection of the most appropriate model, the authors could describe the recurrence interval of historic observations. Is there a temporal pattern? If so, what is the most likely controlling factor for the patter? Can the model generated used to define risks of future flood events? I think is information is presented in the results and discussion section (Figure 3.6), but it is not discussed in length there (their implications) nor in the conclusions.  

Reviewer 2 Report

The authors used the bivariate Copula method to conduct hazard investigation of flood peak and volume in Tarbela dam of Pakistan. Overall the submission, the authors have clearly stated the research purpose and concluded a logistic outcome in your results. I think the paper is worth to be published, and near a well final publication version. I only comment some minor opinions on following for your reference. Thanks.

1. The authors used some well-established methods to conduct the study. An original scientific paper in which relative to applications should emphasize on either the parameters/coefficients determination or scientific findings of adopting site. In your case, more significant explanations should be addressed on, for example, how the characteristics of your parameters/coefficients, or how particular findings in your site? The study may have more discussions instead of just listing statistics results. In your study, return period of 10000 year is calculated. In fact, the value is only meaningful in statistic.

2. To your results (see on Fig. 3.1), the peaks flow revealed single peak in (1) but shown a double peak pattern in volume plot (shown as (2)). The pattern is vague, and more clear clarification is suggested.

3. An unclear expression of Table 3.1, how many events during 1977-2012 were studied. Please specify.

Reviewer 3 Report

This manuscript describes the frequency analysis of the flooding occurred in Indus River of Pakistan using hydrological method to determine the return period and to describe the peak, volume and duration of the flood. 

Due to the scarsity of english, the bad organization of the sections of the manuscript, the not so clear description of the data used, the references very very old except for some of them, I suggest to reorganize the paper and then resubmit it. 

You should put all the matematical formula in appendix in order to make the paper less heavy for the reader, focussing  more the text  on the aim  of the paper.